# In-Vitro Application of Magnetic Hybrid Niosomes: Targeted siRNA-Delivery for Enhanced Breast Cancer Therapy

**DOI:** 10.3390/pharmaceutics13030394

**Published:** 2021-03-16

**Authors:** Viktor Maurer, Selin Altin, Didem Ag Seleci, Ajmal Zarinwall, Bilal Temel, Peter M. Vogt, Sarah Strauß, Frank Stahl, Thomas Scheper, Vesna Bucan, Georg Garnweitner

**Affiliations:** 1Institute for Particle Technology, Technische Universität Braunschweig, 38104 Braunschweig, Germany; v.maurer@tu-braunschweig.de (V.M.); selinaltin91@gmail.com (S.A.); d.ag-seleci@tu-braunschweig.de (D.A.S.); a.zarinwall@tu-braunschweig.de (A.Z.); b.temel@tu-braunschweig.de (B.T.); 2Center of Pharmaceutical Engineering (PVZ), Technische Universität Braunschweig, 38106 Braunschweig, Germany; 3Laboratory for Emerging Nanometrology (LENA), Technische Universität Braunschweig, 38106 Braunschweig, Germany; 4Department of Plastic, Aesthetic, Hand and Reconstructive Surgery, Hannover Medical School, 30625 Hannover, Germany; vogt.peter@mh-hannover.de (P.M.V.); strauss.sarah@mh-hannover.de (S.S.); Bucan.Vesna@mh-hannover.de (V.B.); 5Institute for Technical Chemistry, Leibniz University Hannover, 30167 Hannover, Germany; stahl@iftc.uni-hannover.de (F.S.); scheper@iftc.uni-hannover.de (T.S.)

**Keywords:** gene delivery, siRNA, magnetic targeting, niosomes, hybrid nanoparticles

## Abstract

Even though the administration of chemotherapeutic agents such as erlotinib is clinically established for the treatment of breast cancer, its efficiency and the therapy outcome can be greatly improved using RNA interference (RNAi) mechanisms for a combinational therapy. However, the cellular uptake of bare small interfering RNA (siRNA) is insufficient and its fast degradation in the bloodstream leads to a lacking delivery and no suitable accumulation of siRNA inside the target tissues. To address these problems, non-ionic surfactant vesicles (niosomes) were used as a nanocarrier platform to encapsulate Lifeguard (LFG)-specific siRNA inside the hydrophilic core. A preceding entrapment of superparamagnetic iron-oxide nanoparticles (Fe_x_O_y_-NPs) inside the niosomal bilayer structure was achieved in order to enhance the cellular uptake via an external magnetic manipulation. After verifying a highly effective entrapment of the siRNA, the resulting hybrid niosomes were administered to BT-474 cells in a combinational therapy with either erlotinib or trastuzumab and monitored regarding the induced apoptosis. The obtained results demonstrated that the nanocarrier successfully caused a downregulation of the LFG gene in BT-474 cells, which led to an increased efficacy of the chemotherapeutics compared to plainly added siRNA. Especially the application of an external magnetic field enhanced the internalization of siRNA, therefore increasing the activation of apoptotic signaling pathways. Considering the improved therapy outcome as well as the high encapsulation efficiency, the formulated hybrid niosomes meet the requirements for a cost-effective commercialization and can be considered as a promising candidate for future siRNA delivery agents.

## 1. Introduction

At present, breast cancer is not only the most frequently diagnosed cancer disease in women worldwide but also responsible for most cancer-related deaths [1]. Depending on the type and stage of the tumor, different clinically established therapies such as hormone, radio-, immuno- and chemotherapy or tumor excision as well as a combination of those methods are practiced to treat the potentially lethal disease [2,3]. Especially for hormone receptor- and HER2 (human epidermal growth factor receptor 2)-positive breast cancer, chemotherapy is a commonly used therapeutic method [4,5]. For this type of cancer, various studies have shown that HER2 is responsible for increased activation of a signaling pathway called PI3-K/Akt (phosphoinositide-3-kinase/protein kinase B), which is considered to play an essential role in disease progression [6,7,8,9]. Suppression of the PI3-K/Akt signaling cascade can be achieved by using pharmacologically active substances such as trastuzumab (anti-HER2 monoclonal antibody) or erlotinib (epidermal growth factor inhibitor) and, thus, prevent further tumor cell proliferation [10,11]. Nevertheless, chemotherapeutic treatments still exhibit limitations—for example, side effects in off-target tissues, lack of tumor targeting and especially multi-drug resistance [12,13,14]. Therefore, newly developed treatments are emerging to either replace chemotherapy or enhance its efficiency while lowering potential adverse effects. Considerable attention has been directed to regulate genes responsible for cancer pathology [15,16,17]. For instance, using small interfering RNA (siRNA) as an RNA interference (RNAi) mechanism enables both a mono- and combinational therapy with other therapeutics by silencing specific genes and inhibiting the expression of the respective protein [18,19,20,21]. To date, many studies have been conducted using synthetic siRNA to inhibit genes of interest by the degradation of their messenger RNA transcript in the cell cytoplasm, thereby inhibiting the expression of a specific protein [22,23,24]. For example, it is assumed that the anti-apoptotic protein Lifeguard (LFG, a membrane-bound protein) is increasingly expressed via activation of the Pi3-K/Akt signal cascade and, thus, plays a decisive role in the inhibition of programmed cell death [25,26]. In this case, Bucan et al. showed that combinational therapy consisting of chemotherapeutic downregulation of the Pi3-K/Akt pathway together with a preceding siRNA treatment to minimize the LFG expression level in MCF-7 breast cancer cells led to a cumulative apoptotic outcome with a significantly suppressed cell proliferation and survival [18,27,28,29].

In comparison to anti-cancer drugs, the use of siRNAs in cancer therapy is tremendously beneficial. For instance, siRNA only affects the post-transcriptional phase of gene expression and does not interfere directly with DNA, thus avoiding mutations and teratogenicity risks [24]. Furthermore, siRNA therapy has a high efficacy due to large suppression of gene expression levels with just a few copies [30]. However, safe and effective clinical application of siRNA remains challenging. Since siRNA is negatively charged, administration without a protective layer can lead to non-specific binding with serum proteins and, thus, failed cell admission due to electrostatic repulsion to the negatively charged cell membrane [31]. If the target tissue is not reached, the pharmacological properties of the administered siRNA can potentially cause side effects, for example, by the downregulation of LFG in brain cells after crossing the blood–brain barrier [32]. Moreover, the vulnerable nature of siRNA, the nuclease degradation of siRNA in the blood and the possibility of siRNA removal by renal excretion or macrophages generally impede its delivery to the tumor cells [1,33,34]. Therefore, it is of crucial importance that siRNA is formulated into a suitable nanocarrier system to protect it from the aforementioned barriers and to provide safe and effective delivery into the desired cells and tissues.

So far, a number of nanocarrier systems for therapeutic gene delivery have been described, such as liposomes, chitosan and different cationic nanoparticles [35,36,37,38,39,40,41,42]. Utilizing the leaking vascular system of solid tumors, nanocarriers are able to pass through the vascular barrier. It has been shown that nanoparticles of a certain size ranging up to 200 nm can migrate into tumor cells [43]. Ideal nanocarriers should ensure the criteria of biocompatibility, biodegradability and appropriate physicochemical properties to reach the target cells and tissues [44]. Recently, the first liposome-based siRNA therapeutic was approved by the U.S. Food and Drug Administration (FDA), which underlines the great potential of such RNAi agents [45].

Among nanocarriers, niosomes (non-ionic surfactant vesicles) exhibit great properties to facilitate siRNA delivery [46]. Niosomes show low toxicity and high biocompatibility within biological systems. Moreover, their low cost, simple preparation and remarkable stability make them a notable candidate for potential commercialization. In order to achieve specific functionalities, the hydrophilic core and the hydrophobic bilayer structure can incorporate various low-molecular substances, proteins, DNA/RNA and nanoparticles with a wide range of solubilities [47,48]. If entities with strong magnetic properties (e.g., superparamagnetic iron oxide nanoparticles, referred to as Fe_x_O_y_-NPs) are entrapped, an external magnetic field can be used to achieve a magnetic targeting functionality of the resulting hybrid niosomes [49,50]. Thus, the superparamagnetic structures can be guided to and accumulate at the intended target tissue which consequently lowers concentrations in off-target sites and, hence, significantly improves the therapeutic outcome [51,52,53,54].

Additionally, the niosomal surface can be coated with hydrophilic polymers such as polyethylene glycol (PEG), which is non-toxic, non-immunogenic, non-antigenic as well as highly water-soluble and prolongs the blood circulation time by preventing enzymatic degradation [55]. Due to these favorable properties, niosomes are considered to play a major role in future drug delivery and bear the potential for use in commercially available therapeutics.

Herein, we present the synthesis of novel therapeutic nanocarriers by concurrent encapsulation of Fe_x_O_y_-NPs and LFG gene-silencing siRNA into niosomes (Figure 1). The Fe_x_O_y_-NPs were fabricated by solvothermal decomposition, entrapped inside the hydrophobic bilayer of niosomes (referred to as Fe_x_O_y_/NIO) and used for in vitro magnetic targeting by applying an external magnetic field. Synthetic siRNA, designed to target the LFG gene, was condensed with protamine to achieve an efficient encapsulation into the hydrophilic core of the niosomes through the thin film hydration method. After an in-depth characterization of the resulting hybrid niosomes (referred to as siRNA/Fe_x_O_y_/NIO), the combinational therapy efficacy of the hybrid niosomes together with the anti-tumor drugs trastuzumab and erlotinib was investigated using hormone receptor- and HER2-positive BT-474 breast cancer cells. The increased resulting apoptotic induction caused by downregulation of the LFG gene with and without external magnetic field exposure was analyzed using different apoptosis assays.

## 2. Materials and Methods

### 2.1. Chemicals and Reagents

Benzyl ether (98%), cholesterol (≥99%), 1,2-hexadecanediol (90%), insulin solution (human), iron(III) acetylacetonate (Fe(acac)_3_, ≥99%), non-essential amino acid solution (100×), oleic acid (90%), oleylamine (70%), phosphotungstic acid (99.99%), phosphate-buffered saline (PBS, tablet form), protamine, sodium pyruvate solution, sorbitan monostearate (Span60), SYBR Green I dye, Tris base-acetic acid-EDTA buffer (TAE) buffer (1×) and trypsin were obtained from Sigma-Aldrich Chemie GmbH (Taufkirchen, Germany). Additionally, 1,2-distearoyl-*sn*-glycero-3-phosphoethanolamine-*N*-[maleimide(polyethyleneglycol)-2000] (DSPE-PEG (2000) maleimide) was provided by Avanti (Alabaster, AL, USA). The DNA ladder, loading dye and ethanol (99.5%) were supplied by Thermo Fisher Scientific GmbH (Schwerte, Germany). Nuclease-free water and siRNA (5′-guugcugaguguacucuaa-3′, target sequence of LFG gene) were purchased from Qiagen (Hilden, Germany). Agarose and Roti-Safe Gel Stain were supplied by Roth (Karlsruhe, Germany). Dulbecco’s Minimum Essential Medium (DMEM, supplement-free) was purchased from Gibco Life Technologies (Carlsbad, CA, USA). Fetal calf serum (FCS) was ordered from Biochrom GmbH (Berlin, Germany). BT-474 (isolated from a solid, invasive ductal carcinoma of the breast) cell line was provided from Cell Lines Service (CLS; Eppelheim, Germany). All chemicals were used as received. Unless otherwise stated, ultrapure water was used.

### 2.2. Synthesis of Fe_x_O_y_-NPs

Superparamagnetic Fe_x_O_y_-NPs were synthesized via an adapted solvothermal decomposition method [56,57,58]. Initially, 5 mmol of 1,2-hexadecanediol was mixed with 10 mL benzyl ether and heated to 100 °C for 30 min. Subsequently, 3 mmol each of the stabilizers oleic acid and oleylamine as well as 1 mmol of the molecular precursor Fe(acac)_3_ were added. The reaction mixture was refluxed under nitrogen atmosphere, first at 200 °C for 30 min, followed by 265 °C for 30 min. After cooling down to room temperature, the NPs were washed with ethanol thrice and dispersed in chloroform. Since this synthesis generates iron oxide NPs with contents of magnetite Fe_3_O_4_ and maghemite γ-Fe_2_O_3_, the general term Fe_x_O_y_-NPs was used.

### 2.3. Preparation of siRNA/protamine Polyplex-Loaded Fe_x_O_y_-NIOs

The synthesis of PEG-maleimide-functionalized niosomes was based on the thin film hydration method according to the protocol stated by Ag Seleci et al. [48]. In short, Span60, cholesterol and DSPE-PEG (2000) maleimide were dispersed in 1 mL chloroform (4.95:4.95:0.1 mM). To entrap Fe_x_O_y_-NPs inside the niosome bilayer structure, 2.5 µL of the Fe_x_O_y_-NP suspension (7.0 mg/mL) was added to the lipid mixture. The solvent was evaporated using a rotary evaporation process for 2 h at 38 °C under vacuum (300 mbar), which resulted in the formation of the thin film. Then, 1 mL of siRNA/protamine polyplex suspension, which had been previously prepared by mixing the siRNA and protamine in a 1:3 molar ratio (N/P ratio of 1.5) at room temperature for 30 min, was used to hydrate the lipid film. For the synthesis of plain Fe_x_O_y_/NIO, 1 mL of water was used instead. The resulting mixture was sonicated for 30 min at 65 °C. Subsequently, utilizing a mini extruder set (Avanti Polar Lipids, city, state abbreviation, USA), 15 extrusion cycles were implemented, first using a 0.4-µm and then a 0.1-µm polycarbonate membrane to obtain siRNA/Fe_x_O_y_/NIOs with a particle size around 130 nm and a molar ratio of niosomes:Fe_x_O_y_-NPs:polyplexes of 10:9 × 10^−7^:3.2 × 10^−7^ mM. To remove free protamine and siRNA/protamine polyplexes, the hybrid NP suspension was treated with trypsin for 10 min at 37 °C [59]. Finally, siRNA and trypsin residues were removed by centrifuging thrice using a 100-kDa filter.

### 2.4. Cell Culture

The BT-474 cell line was cultured in Dulbecco’s Modified Eagle Medium and Ham’s F12 (DMEM; Ham’s F12, PAA, Cölbe, Germany) supplemented with 10% FCS, containing 0.1 units/mL bovine insulin (5 mg/mL). Cells were maintained at 37 °C with 5% carbon dioxide in a humidified atmosphere. The medium was changed every 2 to 3 days, and cells were subcultured by treatment with 0.25% Trypsin/0.53 mM ethylenediaminetetraacetic acid (EDTA) solution.

### 2.5. Characterization Methods

For the powder X-ray diffraction (XRD) analysis, an Empyrean series 2 from Malvern Panalytical (Kassel, Germany) with Cu-K_α_ radiation (wavelength λ of 0.154 nm) was used. The Fe_x_O_y_-NPs were dried in an oven, put onto a Si sample holder and measured in the range of 20° to 90° 2θ with a step size of 0.05°. The obtained diffractogram was compared with reference spectra from the Inorganic Crystal Structure Database (ICSD).

The surface chemistry of the Fe_x_O_y_-NPs was analyzed by attenuated total reflectance Fourier-transform infrared spectroscopy (ATR-FT-IR) using a Vertex 70 device from Bruker (Billerica, MA, USA) after drying the sample in an oven overnight.

Transmission electron microscopy (TEM) images were recorded with a Tecnai G² F20 TMP (turbo-molecular pump) from Fei (Hillsboro, OR, USA). Furthermore, 20 µL of Fe_x_O_y_-NPs and Fe_x_O_y_/NIOs each were mixed with 10 µL of a 2% aqueous phosphotungstic acid staining solution and dropped onto a carbon film on a 3.05-mm woven copper net with 300 mesh from Plano GmbH (Wetzlar, Germany).

Dynamic light scattering (DLS) and zeta potential measurements of Fe_x_O_y_-NPs and siRNA/Fe_x_O_y_/NIOs were taken using a Zetasizer Nano ZS and the Zetasizer Nano software (v7.12) from Malvern Panalytical (Kassel, Germany). The measurements were prepared at 23 °C with a 173° backscattering setup. Previous dilution of the samples by a factor of 10^4^–10^5^ minimized fluorescence. To obtain the hydrodynamic diameter, the data evaluation was based on the modi of the respective intensity distributions. The zeta potentials were obtained using a capillary zeta cuvette (DTS1070C, Malvern Panalytical).

The magnetic behavior and saturation magnetization of Fe_x_O_y_-NPs and Fe_x_O_y_/NIOs was examined via a superconducting quantum interference device (SQUID) using the MPMS-5S instrument from Quantum Design (Darmstadt, Germany). The samples were dried analogously to the FT-IR preparation.

Agarose gel electrophoresis was performed to investigate the siRNA encapsulation efficiency. Free siRNA, siRNA/Fe_x_O_y_/NIOs as well as the washing solution were loaded to a 1.5% agarose gel prepared in 1 × TAE buffer and stained with 5.0 µL/100 mL Roti-Safe Gel Stain. A Thermo EC electrophoresis device (Thermo Fisher Scientific, Waltham, MA, USA) was utilized to run the gel at 100 V for 60 min. Subsequently, an image was taken via the INTAS UV documentation system (Intas Science Imaging Instruments, Göttingen, Germany).

The SYBR Green I dye was applied to free siRNA and siRNA/Fe_x_O_y_/NIOs for the quantification of encapsulated siRNA according to Saxena et al. [60]. The fluorescence intensities of samples were measured at 650 nm using a NanoDrop 3300 fluorospectrometer from Thermo Fisher Scientific (Waltham, MA, USA). The concentration of the encapsulated siRNA, conc. (encapsulated siRNA), can be calculated by putting the values for the concentration of free siRNA, conc. (free siRNA), the detected relative fluorescence units (RFU) for encapsulated siRNA and the RFU of free siRNA in the following formula:(1)$$\mathrm{conc}.\;\left(\mathrm{encapsulated}\;\mathrm{siRNA}\right)\;=\;\frac{\mathrm{conc}.\;\left(\mathrm{free}\;\mathrm{siRNA}\right)\cdot \;\mathrm{RFU}\;\left(\mathrm{encapsulated}\;\mathrm{siRNA}\right)}{\mathrm{RFU}\;\left(\mathrm{free}\;\mathrm{siRNA}\right)}$$

Assessment of the metabolic activity of viable cells according to their caspase 3/7 activity as well as the intracellular calcium mobilization was carried out using the Apo-One Homogeneous Caspase-3/7 assay (Promega, Madison, WI, USA) according to the manufacturer’s instructions and the Screen Quest™ Calbryte-520 Assay Kit, respectively. Briefly, BT-474 breast cancer cells were seeded (10^4^/well) in two different 96-well plates for 24 h and, afterwards, treated with the respective sample (siRNA samples always contained 50 µg/mL siRNA), followed by 48 h of incubation with 5.0% CO_2_ at 37 °C. The impact of an external magnetic field (neodymium magnet with a magnetic field of 1.3 T) on the nanoparticle uptake and therapeutic effect was investigated by placing a permanent magnet under one 96-well plate for 15 min before the aforementioned 48-h incubation (samples referred to as M+). After this pretreatment, cells were incubated with 1 mg/mL of trastuzumab (Herceptin^®^, Roche, Basel, Switzerland) or erlotinib (Tarceva^®^, Roche, Basel, Switzerland) for 4 h, respectively. The caspase 3/7 activity was investigated by putting 100 µL of caspase reagent into each well of the 96-well plate for 2 h at room temperature. Caspase 3/7 activation was estimated from sample fluorescence at the excitation wavelength of 492 nm and the emission wavelength of 521 nm using the fluorescence plate reader Tecan GENios (Tecan Schweiz AB, Zurich, Switzerland). Screening for free calcium channels of fragile cells was carried out by adding 100 µL of the Calbryte dye-loading solution to each well of a 96-well plate and incubating the cells for 60 min. Finally, the samples of both plates were analyzed using the microplate reader at excitation/emission 492/521 nm. The resulting values are directly proportional to the amount of apoptosis. To compare the significance of the results, the values of the samples containing siRNA/Fe_x_O_y_/NIOs + anti-tumor agent and the values of the respective anti-tumor agent were analyzed using the Student’s *t*-test.

## 3. Results

### 3.1. Preparation of Fe_x_O_y_-NPs

For incorporation into the niosomal bilayer, Fe_x_O_y_-NPs should not exceed a particle size of 8 nm and their surface must be capped with hydrophobic functional groups. Therefore, the solvothermal decomposition method in the presence of oleic acid and oleylamine, which allows the preparation of hydrophobic and crystalline Fe_x_O_y_-NPs with an adjustable particle size, was implemented. The synthesized Fe_x_O_y_-NPs revealed a spherical shape and a particle size ranging between 3 and 8 nm (Figure 2a). DLS measurements unveiled a hydrodynamic diameter of approx. 6.7 nm in hexane, with the Fe_x_O_y_-NPs being equally stable in other non-polar solvents such as chloroform. The X-ray diffractogram displays a crystalline mixed phase of magnetite and maghemite (ICSD 98-015-8714 and ICSD 98-007-9196), which is a typical result for nanoscale iron oxide particles [61,62].

Since the cubic structure and the lattice parameters of magnetite and maghemite are quite similar, a more accurate determination of the iron oxide composition is not possible via XRD. Applying the Debye–Scherrer equation to the highest intensity reflection at 36.6° with an FWHM (full width at half maximum) of 1.5° and Ks (dimensionless shape factor for spherical particles) of 0.9 leads to a crystallite size of 5.8 nm, which is in good agreement with the particle sizes obtained by TEM and DLS. Furthermore, the FT-IR spectrum of the Fe_x_O_y_-NPs shows alkyl group-based peaks at 2916 and 2842 (symmetric and asymmetric CH_2_ stretching modes), 1522 and 1400 cm^−1^ (NH_2_ scissoring mode as well as asymmetric and symmetric COO^–^ stretching) which confirm the presence of hydrophobic stabilizing agents on the Fe_x_O_y_-NP surface (Figure 2d) [63,64].

### 3.2. Synthesis of siRNA- and Fe_x_O_y_-Loaded Niosomes

After successful synthesis of the Fe_x_O_y_-NPs, the siRNA/Fe_x_O_y_/NIO hybrid nanoparticles were prepared and analyzed by investigating their resulting particle properties. As displayed on TEM images, the encapsulation of Fe_x_O_y_-NPs led to the formation of spherical Fe_x_O_y_-NP clusters with a niosome-specific size of approx. 100 nm (Figure 3a).

DLS measurements demonstrate a niosome-typical hydrodynamic diameter of around 145 nm for the prepared Fe_x_O_y_/NIO and a small reduction in the vesicle size to 127 nm when a concurrent encapsulation of siRNA/protamine polyplexes was carried out (Figure 3b). This was similarly reported in another study [65] and is considered to be in the optimum range for nanocarriers for in vivo tumor accumulation [43]. The stability of siRNA/Fe_x_O_y_/NIOs was tested via DLS analysis and no changes were observed in the size and PDI (polydispersity index) values after a two-week storage at 4 °C in the dark (data not shown). Moreover, magnetic measurements revealed a superparamagnetic behavior for the Fe_x_O_y_-NPs with saturation magnetization of 42.1 Am²/kg (Figure 3c), which is consistent with previous findings [66,67,68].

After incorporation of Fe_x_O_y_-NPs into the niosomes, the measurement of the resulting material still did not show a magnetic hysteresis, thereby proving superparamagnetic properties with a decreased saturation magnetization of 11.6 Am²/kg due to the organic matrix [65,69]. In addition, zeta potential measurements were carried out to evaluate the efficacy of the trypsin treatment (+TT) after siRNA encapsulation (Figure 3d). The siRNA entrapment procedure resulted in a shift of the zeta potential from −41.1 to −37.9 mV due to the attachment of positively charged polyplexes onto the negatively charged niosome surface. The decrease in the zeta potential to 43.3 mV after trypsin treatment indicates the removal of the polyplexes [59].

To verify the successful encapsulation of siRNA/protamine polyplexes inside the Fe_x_O_y_/NIOs, agarose gel electrophoresis with RNA-specific dye staining was performed (Figure 4a). The concentration for the siRNA encapsulation was adjusted according to the detection limit of the dye. In comparison to the free siRNA in well (2), siRNA in the siRNA/Fe_x_O_y_/NIO-loaded well (3) did not migrate since it was incorporated inside the niosome structure and, hence, sterically hindered. No band could be detected in the pocket containing the wash solution after trypsin treatment, implying the presence of only a negligible amount of siRNA in the supernatant after purification, thereby signifying a high siRNA encapsulation efficiency.

To further evaluate the entrapment of siRNA, fluorescence measurements of free siRNA (conc. (free siRNA) = 5.0 µM), siRNA/Fe_x_O_y_/NIO (initial conc. used for encapsulation = 3.5 µM) and the wash solution were carried out. The detected RFU values were 135.7 (free siRNA), 94.0 (encapsulated siRNA) and 0.0 (wash solution) (Figure 4b). By using Equation (1), a concentration of 3.46 µM for the encapsulated siRNA was calculated, which indicates a 99.0% encapsulation efficiency. This finding confirms the gel electrophoresis results and proves a highly effective entrapment of siRNA/protamine polyplexes into niosomes.

### 3.3. Combinational Therapy and In Vitro Cytotoxicity

Finally, we investigated whether the nanocarrier-assisted siRNA delivery can effectively regulate the expression level of LFG in LFG-overexpressing BT-474 breast cancer cells, causing a significantly enhanced drug efficacy. An enhanced apoptotic effect after transfection of anti-LFG siRNA in a combinational therapy with different anti-tumor drugs compared to a control siRNA was already proven recently [18]. As therapeutic agents, trastuzumab and erlotinib were selected, which are both directed against tyrosine kinase receptors and inhibit pathways that control cell proliferation and overall progression of metastatic breast cancer in women [70,71]. To evaluate the in vitro efficacy of a combinational therapy consisting of trastuzumab/erlotinib and siRNA/Fe_x_O_y_/NIO, the apoptosis-inducing activity was investigated by analyzing different cellular factors such as caspase activity and intracellular Ca^2+^ concentration levels. Caspase 3/7 cleavage activity was detected using the Apo-ONE assay, and the Ca^2+^ concentration levels were evaluated by utilizing the Calbryte-520 Assay Kit, with the detected fluorescent signals being proportional to the induced apoptosis (Figure 5).

After administration of siRNA/Fe_x_O_y_/NIO and erlotinib, the BT-474 cells showed a significantly increased amount of caspase 3/7 cleavage activity (1.86 × 10^4^) compared to administration of the drug with (1.58 × 10^4^) and without (1.34 × 10^4^) a preceding siRNA addition (Figure 5a, *p* < 0.05). Since the induced apoptosis of the plain siRNA/Fe_x_O_y_/NIO is almost negligible (1.46 × 10^4^ compared to free siRNA, 1.51 × 10^4^), the enhanced caspase 3/7 cleavage activity can be attributed to improved siRNA delivery due to the niosomes. In fact, the addition of erlotinib with prior siRNA incubation (1.58 × 10^4^) does not result in a significant increase in caspase 3/7 cleavage activity compared to the administration of plain siRNA (1.51 × 10^4^). In the case of trastuzumab, a preceding administration of siRNA/Fe_x_O_y_/NIO leads to a similar apoptotic effect (1.94 × 10^4^) as the administration of plain siRNA + trastuzumab (2.01 × 10^4^), which, likewise, proves the effective nanocarrier function of the niosomes. Similarly to erlotinib, trastuzumab itself does not result in a higher fluorescence when administered without siRNA treatment (1.52 × 10^4^, *p* < 0.05).

The same trend can be seen when evaluating the cellular Ca^2+^ levels (Figure 5b): application of siRNA/Fe_x_O_y_/NIO in combination with erlotinib results in a higher occurrence of apoptosis (2.66 × 10^4^) due to higher intracellular Ca^2+^ concentration levels than the administration of the drug with (2.53 × 10^4^) or without (2.41 × 10^4^) a preceding siRNA addition (*p* < 0.05). Furthermore, this effect was enhanced by employing an external magnetic field after administration of siRNA/Fe_x_O_y_/NIO (2.90 × 10^4^), which indicates a higher cellular internalization and, therefore, a successful magnetic targeting capability. This result was elucidated by comparing the enhanced signal on fluorescent images of siRNA/Fe_x_O_y_/NIO with erlotinib in contrast to the other samples (Figure 5c). In addition, combinational treatment with trastuzumab accentuates the successful nanocarrier properties of niosomes; even though the siRNA/Fe_x_O_y_/NIOs together with trastuzumab exhibit a smaller relative fluorescent activity (2.53 × 10^4^), the influence of an external magnetic field enhances the apoptotic effect (2.86 × 10^4^) almost to the stage of the treatment with plain siRNA and trastuzumab (2.95 × 10^4^, Figure 5b). These results not only confirm the successful encapsulation of siRNA and Fe_x_O_y_-NPs into the niosomes but also the applicability of the resulting therapeutic niosomes in order to achieve downregulation of the LFG gene and, thereby, an increase in the apoptosis rate after subsequent treatment with different drugs.

## 4. Discussion

It has been suggested that the usage of siRNA as a therapeutic can be vastly advantageous for cancer treatment. However, optimizing the safe delivery of siRNA and maintaining its activity must be ensured for clinical applications. In this study, for the first time, superparamagnetic Fe_x_O_y_-NPs and niosomes were used to obtain a successful and enhanced siRNA delivery into breast cancer cells to downregulate the LFG gene and, hence, increase the therapeutic efficiency of different drugs.

In the first part of the presented study, superparamagnetic Fe_x_O_y_-NPs were prepared using a solvothermal decomposition. Since the NPs were in the size range of approximately 3–7 nm, spherically shaped and had a hydrophobic nature, a post-synthetic encapsulation inside the bilayer structure of niosomes was achieved. Furthermore, a concurrent incorporation of siRNA/protamine polyplexes could be realized by using a highly effective entrapment procedure with an encapsulation efficiency of approx. 99%. The resulting hybrid niosomes showed a hydrodynamic diameter of approx. 145 nm and superparamagnetic properties. Subsequent trypsin treatment caused the removal of polyplexes attached to the niosomal surface, ensuring a zeta potential similar to plain Fe_x_O_y_/NIO. The affirmed efficacy of the implemented encapsulation together with the generally low-cost synthesis of niosomes will be of major importance for the commercialization of siRNA-based therapeutics, since cost-effective production is a crucial aspect for the establishment of novel clinical therapies.

Afterwards, a silencing of the LFG gene and, hence, downregulation of the protein expression was achieved after administration of the siRNA/Fe_x_O_y_/NIOs to BT-474 breast cancer cells. Subsequent addition of trastuzumab or erlotinib resulted in significantly higher caspase 3/7 cleavage activity as well as increased intracellular Ca^2+^ concentration levels, which undoubtedly indicates the successful activation of the Pi3-K/Akt apoptotic pathway. These findings suggest that the combinational therapy consisting of chemotherapeutic agents and LFG-specific siRNA/Fe_x_O_y_/NIOs led to a cumulative apoptotic effect. By applying an external magnetic field after addition of siRNA/Fe_x_O_y_/NIOs, the magnetic targeting capability remarkably increased the apoptosis efficiency of the combinational therapy in contrast to the plain drugs. Besides providing evidence showing that utilizing a gene-silencing substance in combination with a chemotherapeutic agent for breast cancer is a more efficient treatment in comparison with conventional chemotherapy, we also presented an effective nanoparticulate delivery system by protecting siRNA via encapsulation in a niosomal formulation and enhanced the combinational therapy outcome by use of magnetic particle manipulation.

Whilst an apoptotic enhancement was achieved, we expect that the specificity towards breast cancer cells might be further enhanced by conjugating targeting ligands onto the siRNA/Fe_x_O_y_/NIO surface, which will be subject to future investigations. Moreover, in vivo experiments with a special focus on the safe and selective delivery of therapeutic nanocarriers will be necessary to validate the in vitro results as well as investigate potential side effects. In essence, the niosomal formulation together with the results reported in this study have the potential to provide a novel platform for a new generation of commercially available and highly efficient siRNA-based therapies.

## Figures and Tables

**Figure 1 pharmaceutics-13-00394-f001:**
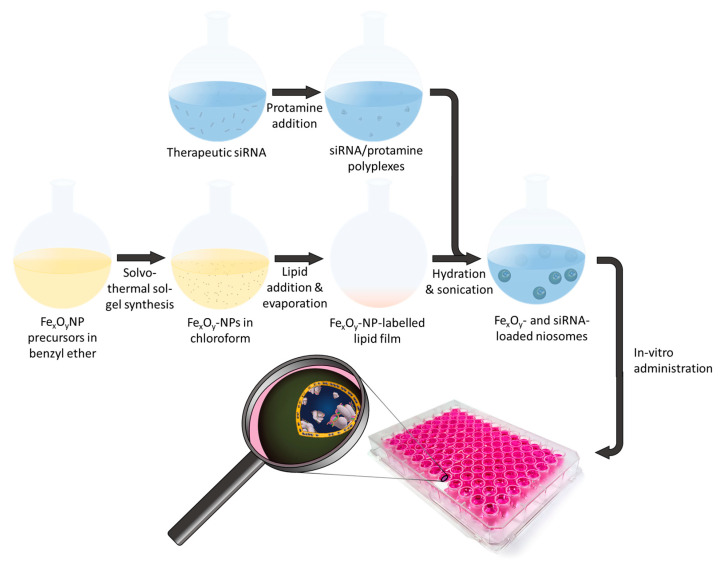
Schematic illustration of the hybrid nanoparticle (NP) formation.

**Figure 2 pharmaceutics-13-00394-f002:**
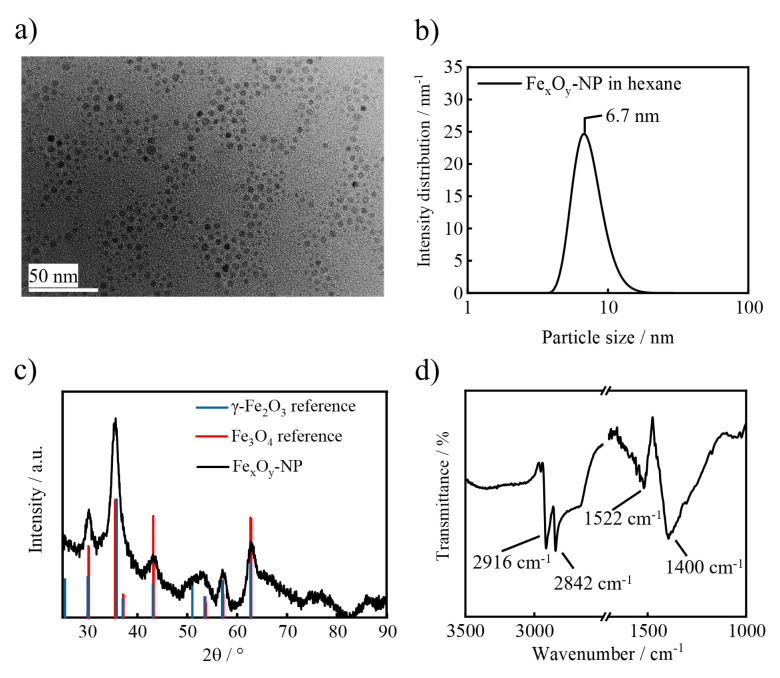
(**a**) Representative TEM image, (**b**) dynamic light scattering (DLS) result, (**c**) X-ray diffraction (XRD) diffractogram with a reference pattern of maghemite γ-Fe_2_O_3_ and magnetite Fe_3_O_4_ and (**d**) FT-IR spectrum of Fe_x_O_y_-NPs.

**Figure 3 pharmaceutics-13-00394-f003:**
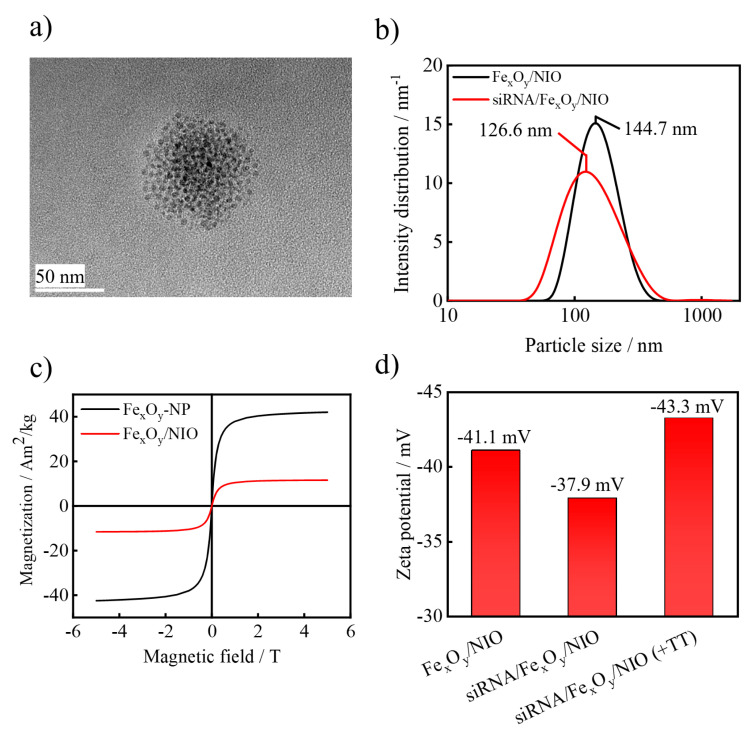
(**a**) Representative TEM image of Fe_x_O_y_/NIO (measured at 200 kV), (**b**) DLS result of plain Fe_x_O_y_/NIOs and siRNA/Fe_x_O_y_/NIOs, (**c**) magnetic behavior of the synthesized Fe_x_O_y_-NPs and Fe_x_O_y_/NIOs at 300 K determined by a superconducting quantum interference device (SQUID) magnetometer as well as (**d**) zeta potential measurements of plain Fe_x_O_y_/NIOs, siRNA/Fe_x_O_y_/NIOs and siRNA/Fe_x_O_y_/NIOs after trypsin treatment (+TT).

**Figure 4 pharmaceutics-13-00394-f004:**
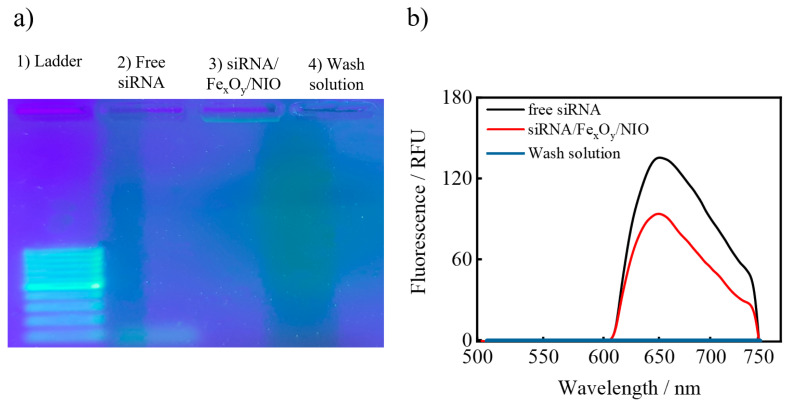
(**a**) Stained agarose gel after electrophoresis of the (1) ladder, (2) free siRNA, (3) siRNA/Fe_x_O_y_/NIOs and (4) wash solution; (**b**) fluorescence spectrum of SYBR Green I-stained free siRNA, siRNA/Fe_x_O_y_/NIO and the wash solution.

**Figure 5 pharmaceutics-13-00394-f005:**
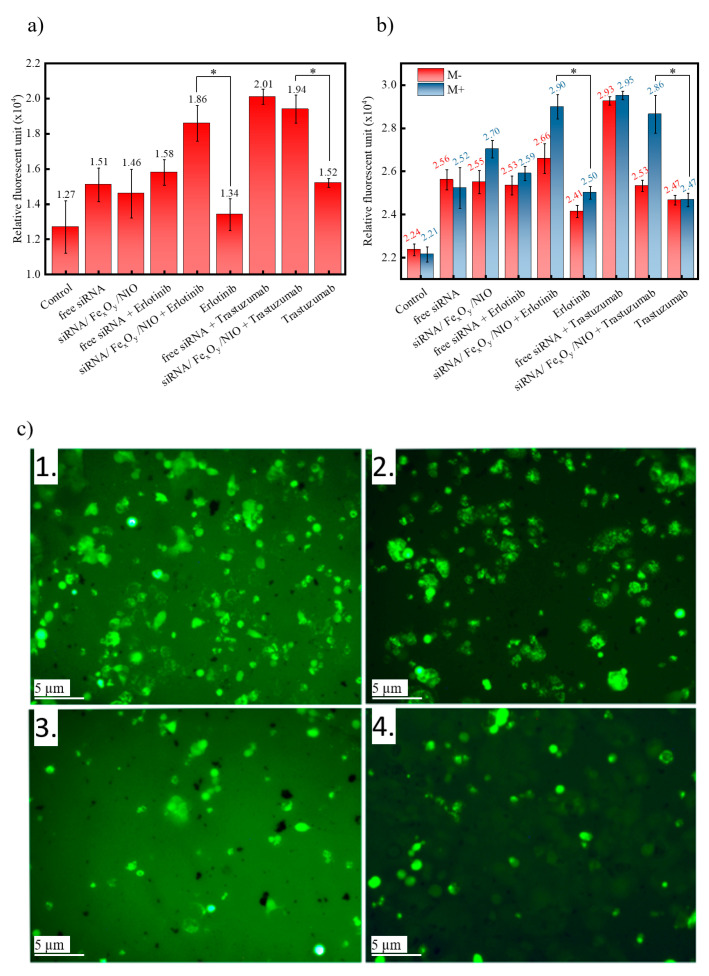
Apoptotic effect determined via (**a**) caspase 3/7 cleavage activity (without magnetic treatment) and (**b**) intracellular Ca^2+^ concentration of free siRNA, siRNA/Fe_x_O_y_/NIO, free siRNA with anti-tumor agent, siRNA/Fe_x_O_y_/NIO with anti-tumor agent and free anti-tumor agent with (M+) and without (M-) magnetic treatment (* = the values of the samples containing siRNA/Fe_x_O_y_/NIO + anti-tumor agent and the value of the respective anti-tumor agent were analyzed using a *t*-test, with all data showing *p* < 0.05 considered significant); (**c**) representative fluorescent images (using the Calbryte-520 Assay Kit) showing occurring apoptosis (light green) of BT-474 cells after administration of (**1**) siRNA/Fe_x_O_y_/NIO with erlotinib, (**2**) free siRNA with erlotinib, (**3**) erlotinib and (**4**) control cells and a subsequent magnetic treatment.

## Data Availability

Data are available upon request.

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
