# Peer review of "In-Vitro Application of Magnetic Hybrid Niosomes: Targeted siRNA-Delivery for Enhanced Breast Cancer Therapy"

_pharmaceutics, 2021, doi:10.3390/pharmaceutics13030394_

Round 1

Reviewer 1 Report

Garnweitner and coworkers reported a cocktail formulation for siRNA delivery based on IONP, protamine, and lipids. The siRNA is chosen for targeting the LFG gene, which leads to cytotoxicity after successful transfection of siRNA. The combination between the non-ionic surfactant vesicles and erlotinib/ trastuzumab leads to enhanced apoptosis. It is a brief and well-organized work and should be suitable for acceptance after addressing a few comments. 

  1. One of the most critical data is missing from the manuscript is the control siRNA, for example in Figure 5a,b. All these evaluations (9 different groups) should be done with the control siRNA. This is necessary because whether RNA interference is the mechanism of action is not clearly demonstrated. As the readout is toxicity, it is hard to perform qPCR for mRNA quantification. 
  2. When co-treating the cells with either erlotinib or trastuzumab, what is the rationale that the combination could lead to an enhanced apoptotic effect, considering there is no physical or chemical interaction between them and the niosomes? Please add the discussion in the manuscript.
  3. In Figure 3, it would be necessary to provide the ratio among each component for the DLS and zeta potential characterization. Similarly, Figure 4a needs to be labeled with the N/P ratio in each lane if applicable. 
  4. Figure 5a, b needs to be analyzed by Student's t-test. Figure 5c is missing scale bars. 
  5. Line 175, what is the quantitative value of the magnetic field applied to the experiment?
  6. A few references on RNA delivery using noncationic platforms should be cited: DOI: 10.1002/adtp.201900206; DOI: 10.1038/s41551-018-0214-1; DOI: 10.1021/acs.biomac.8b01321; DOI: 10.1016/j.biomaterials.2018.02.007. 

Reviewer 2 Report

The manuscript entitled “In-vitro Application of Magnetic Hhybrid Niosomes: Targeted siRNA-Delivery for Enhanced Breast Cancer Therapy” relies on the development and characterization of FeO-Nps loaded niosomes for the delivery Lifeguard (LFG)-specific siRNA and the evaluation of their efficacy in breast cancer therapy in combination with trastuzumab and erlotinib. Here there some suggestions comments that should be addressed.

  • Line 40: Nowadays, chemotherapy represent a mainstay treatment strategy in breast cancer, especially in TNBC and in advanced stages of the diseases in combination with surgery when possible. However, in early stages, and in certain types of tumors hormone-therapy, radiotherapy and immunotherapy are also used and represent good treatment options. I suggest clarifying this point as the paper is focused on hormone receptor positive breast cancer.
  • I suggest including more information in the introduction about trastuzumab and erlotinib, are they are use in combination studies. These drugs are not used in all breast cancer types. Moreover, I suggest focusing on hormone receptor positive breast tumors.
  • In methods section: I suggest reorganizing it. I mean, characterization subsection should be clearer, first the studies in FeO-Nps, then with niosomes, following the organization described in results. Moreover, cytotoxicity studies should be explained at the end, following result organization.
  • Line 257: Please, review this sentence. Does number 28 mean a reference?
  • Equation 1 should be described in methods.
  • Did the authors develop siRNA single loaded niosomes as control?
  • Did the authors evaluate the stability of developed nanoplatform?
  • There are many double parentheses in the text. I suggest reviewing the entire manuscript.

Round 2

Reviewer 2 Report

The manuscript entitled "In-vitro application of magnetic hybrid niosomes: targeted siRNA-delivery for enhanced breast cancer therapy" is suatible for pharmaceutics. The authors have considerably improved the manuscript by adressing all the comments and suggestions. Consequently, the manuscript deserves to be published in the current, revised, form.